# A Method to Facilitate the Regeneration of Human Resources: A Sustainability Perspective

Ningning Chen, Xinan Zhao *, Baorong Guo  and Chenxi Sun

School of Business Administration, Northeastern University, Shenyang 110167, China; 1810446@stu.neu.edu.cn (N.C.); gbr.152@163.com (B.G.); 1710445@stu.neu.edu.cn (C.S.)
* Correspondence: xnzhao@mail.neu.edu.cn

**Abstract:** Within the sustainable development framework, organizations are tasked with creating strategies that ensure the enduring provision of value through human capital for the future. Our study emphasizes employee development and training, adopting a people-centric approach aligned with sustainability principles. By leveraging techniques for the identification of Characteristics of Individual Strengths (CIS), Agent Evaluation, and composite decision making, we introduce a novel approach to formulating personalized employee training strategies. This approach is structured around three pivotal steps: identifying CIS, assessing employee roles within the organization based on CIS, and analyzing training strategies. Demonstrated through illustrative examples, our method validates its applicability in real-world settings. This research provides organizations with an innovative pathway for effectively fostering employee skills and securing a steady influx of high-quality, diverse talent.

**Keywords:** human resource regeneration; sustainability; Characteristics of Individual Strengths; people-centered; methodology; employee training strategies

## 1. Introduction

Guided by the traditional emphasis on economic efficiency, organizations have historically favored cost-minimization strategies when deploying and planning their human resources, such as the preference for reuse over development [1–4], at the expense of their resource bases' stability and sustainability [5]. The advent of Sustainable Human Resource Management (SHRM) marks a paradigm shift towards practices that ensure human resource strategies and activities are both current and future-oriented, balancing immediate needs with sustainable growth. This new focus encourages managers to integrate economic and sustainable development goals through strategic human resources initiatives, prompting a reevaluation of the role of human resource development in achieving long-term sustainability [6]. Under this paradigm, managers have started to explore how to reconcile economic impacts with sustainable development through effective human resources practices and reassess the need for human resource development from a sustainable strategic perspective [7]. Employee development and training, as direct and effective measures for human resources regeneration, have been widely implemented, offering two key assurances for sustainability: firstly, ensuring the continuous acquisition of high-quality labor within the organization [8], and secondly, ensuring the re-creation of labor value and benefit output [9]. Consequently, the quest to refine employee development and training methods continues to be a vital area of inquiry for scholars and managers.

In China, the strategic planning for organizational talent development is conducted periodically, underscored by the "National Medium and Long-term Talent Development Plan Outline." This policy underscores the enhancement of talent cultivation mechanisms, promoting both specialized guidance and holistic development to sustain organizational competitiveness [10]. Within this context, scholars have delved into personnel training

methodologies. Lin [11] discussed the critical role of comprehensive employee development in achieving corporate sustainability and introduced an innovative mechanism for motivating all-around improvement, including capability enhancement and ideological coaching. Lee [12] investigated strategies to bolster employee skills, focusing on training effectiveness. Lin [13], drawing on the principle of resource scarcity, advocated for a shift in organizational focus from addressing weaknesses to accentuating strengths, facilitating superior performance at both individual and organizational levels. While these approaches have shown benefits, they also present significant limitations, notably overlooking employee needs—the core human resource [14]—and underestimating individual variability [15], thus leading to a pronounced unilateral effect in management and affecting the efficacy of development initiatives. Adopting a holistic strategy for sustainable human resources, numerous scholars [16–18] concur that people-centered development is an indispensable element of sustainable growth. This consensus underscores that all strategies for human resource regeneration should be centered around the specific needs of employees. Our study, rooted in this viewpoint, is dedicated to devising a methodology for crafting employee training strategies. This approach is designed to empower organizations to precisely steer employee endeavors towards skill enhancement and strategic goals. The objective is to promote the effective regeneration of human resources, simultaneously reducing the discord between economic gains and the enduring availability of skilled talent.

The remaining sections of this study are organized as follows: Section 2 introduces the framework for our research methodology. Section 3 delineates the methodology for enhancing human resource regeneration, detailing the selection process for employee training strategies. It encompasses the identification of Characteristics of Individual Strengths (CIS), assessing employee roles within the organization based on CIS, and analyzing training strategies. Section 4 validates the proposed method through illustrative examples. Finally, Section 5 concludes the study with a summary.

## 2. Framework for Methodological Research

The decision-making method for employee training strategies is designed to assist staff in charting paths for future enhancement and development. It is essential to determine the management philosophy adhered to during the process of method design, as this dictates whether the organization can enable precise evaluation of employee capabilities and the crafting of compatible training trajectories. Our approach, grounded in sustainability, places the employee at the forefront, valuing their unique contributions and diverse developmental outcomes. This employee-centric view facilitates the meticulous identification of training needs and the creation of customized development plans, thus ensuring a pipeline of high-quality, sustainable talent for the organization. The realization of these goals is supported by technologies for identifying CIS and relevant assessment methodologies, the specific steps of method construction are as follows:

Definition and identification of CIS: Every individual harbors unique strengths that maximize their value. Define it and calculate it using the ideal point method for subsequent application in assessing employee development status.

Role's assessment of employee based on CIS: Acknowledging the variance in employees' characteristics and developmental phases [19,20], it is imperative to scientifically assess their organizational positioning. Unlike traditional assessment methods that utilize uniform criteria, our approach emphasizes individual strengths, offering a nuanced understanding of each employee's role and potential within the organization. This process involves two key assessments: Democratic Agent Evaluation (DAE) for a comprehensive ranking of employees based on concentrated strengths and Individual Agent Evaluation (IAE) to gauge their uniqueness and irreplaceability.

Strategy analysis for employee training: Leveraging the insights gained from the dual-dimensional assessment, employees are categorized based on their organizational roles and developmental phase. This classification informs the formulation of targeted development strategies tailored to the specific attributes of each category.

Through these steps, we will establish a strategic selection process for employee training and demonstrate its applicability through specific illustrative examples. It is noteworthy that our research is grounded in a people-centered philosophy, focusing on model construction and solution techniques. Since the use of these techniques relies on data, which is a quantification of the evaluated subjects' actual performance. Thus, the accuracy and validity of the data significantly impact the assessed results. Nonetheless, due to the focus of our study, an in-depth discussion on data quality is beyond its scope.

## 3. Development of Selection Methods for Employee Training Strategies

### 3.1. Definition and Identification of CIS

Within organizational operations, employees have developed distinct behavioral preferences and strengths through routine work activities [21]. For example, in a team, some members excel in their professional skills, while others may shine in areas such as leadership and management. This diversity in skill sets contributes to an organization's vitality and heterogeneity. The emergence of such diversity is shaped by organizational incentives and individual intrinsic factors [22]. However, conventional evaluation methods often overlook these unique individual traits. For instance, in evaluation systems favoring technical expertise, the abilities of employees skilled in communication and expression may be under-recognized. Each individual has a distinct strengths assessment framework, which, if utilized correctly, can maximize the expression of their value. This framework, known as Characteristics of Individual Strengths (CIS), is key in recognizing and maximizing individual potential. Specifically, it refers to an essential structure that corresponds to the indicator dimensions and can best reflect individual value within a certain range of indicator content and under a certain value concept [23]. Since the introduction of CIS, its innovative concept has led numerous scholars to expand a variety of evaluation methods, including performance assessment, recruitment mechanisms, and incentive systems. These methods have been applied across multiple fields such as strategic management, organizational motivation, and technological innovation [24–26]. For instance, Huang et al. applied CIS techniques to the development of new research teams in universities [27].

CIS is cultivated through extensive learning and practical experience. Analysis and assessment rooted in CIS offer valuable insights for individuals [28]. Essentially, if organizations align performance evaluations and development planning with each individual's CIS, they can comprehensively acknowledge employees' contributions while refining and optimizing their developmental trajectories. This personalized evaluation and development approach serves as a catalyst for unlocking employees' latent potential, elevating job satisfaction [29,30], and ultimately fostering the effective development of human resources and enhancing value creation for the organization. Realizing these objectives necessitates effective assistance from pertinent technical methods, including CIS identification. This approach employs pertinent mathematical methods to compute a set of fundamental structures that best encapsulate individual value within the predefined assessment framework, relying on the actual performance of each individual. The identification process involves the following steps:

Step 1: Establish a bespoke set of indicators that align with the organization's value propositions to evaluate employees' abilities, performance, or achievements.

Step 2: Collect and standardize each employee's performance data based on these indicators.

Step 3: Define a comprehensive evaluation model. In this study, the Ideal Point Method is exemplified (though other models are viable) to construct a CIS-solving model. Let us

consider employees $i$ and indicators $j$. The CIS for employee $i$ is represented as $w_i{}^*$, $w_i^* = \left(w_{i1}^*, w_{i2}^*, \cdots w_{im}^*\right)^\tau, i = 1, 2, \cdots n$, calculated using the following identification formula:

$$
\begin{aligned}
\min_w d^2(f_i, f^*) &= \sum_{j=1}^m w_{ij}^2 \left(f_{ij} - f_j^*\right)^2 \\
s.t. \sum_{j=1}^m w_{ij} &= 1 \\
w_{ij} \geq 0; \; j &= 1, 2, \cdots m; i = 1, 2, \cdots n.
\end{aligned}
\tag{1}
$$

In this formula, $f_i = (f_{i1}, f_{i1}, \cdots, f_{im})$ represents the standardized performance value of an employee, $f^* = (f_1^*, f_2^*, \cdots, f_m^*)$ denotes the ideal performance outcome, $w_i^* = \left(w_{i1}^*, w_{i2}^*, \cdots, w_{im}^*\right)^\tau$ signifies the value parameter structure for employee $i$, and $d(f_i, f^*)$ is the distance between $f_i$ and $f^*$ under $w_i$. Minimizing $d(f_i, f^*)$ indicates high recognition of an employee's performance. At this point, the value parameters can be considered as CIS.

The optimal value for each indicator, $f_j^*$, is set by the organization and can be derived from various methods, such as theoretical bests, historical peaks, future expectations, or decision-maker specifications. Typically set higher than actual performance ($f_j^* > f_{ij}$), it motivates individual progress. The calculation formula is as follows:

$$
\begin{aligned}
w_{ij}^* &= 1 / \left\{ \left(f_j^* - f_{ij}\right)^2 \sum_{j=1}^m \frac{1}{\left(f_j^* - f_{ij}\right)^2} \right\} \\
j &= 1, 2, \cdots m; \; i = 1, 2, \cdots n.
\end{aligned}
\tag{2}
$$

when an employee's performance on a specific indicator matches the optimal value ($f_j^* = f_{ij}$), the sum of all optimal value parameters equals 1, with each parameter distributed evenly. The remaining value parameters are set to 0. Firstly, it is essential to differentiate CIS from mere weighting, despite their similarities in expression; secondly, CIS reflects relative strengths compared to oneself and does not imply absolute superiority on corresponding indicators.

### 3.2. Assessment of Employee Roles within the Organization Based on CIS

3.2.1. Comprehensive Ranking Assessment

In organizational contexts, the ranking of employees based on performance and capabilities typically hinges on a specific evaluation mechanism. When applied to the same employee cohort, varied evaluation methods and guiding principles may yield disparate results [31,32]. Selecting the most fitting evaluation method within the given theoretical framework necessitates consideration of decision makers' intentions and evaluation goals [33]. Commonly, evaluation mechanisms include: an assessment indicator system representing the content to be evaluated, employees' performances on these indicators, a weighting structure denoting decision makers' priorities, and the chosen evaluation method [34]. These elements collectively influence employee rankings. This study, however, focuses not on the comprehensiveness of the indicators or the exactitude of the performance assessments but on the guiding impacts of the evaluation results. Therefore, the structure of the evaluation vector, particularly weight determination, is of significant importance in this study.

This study's core premise is the recognition and utilization of individual strengths for personalized analysis and development strategy formulation. When evaluating employees, understanding and valuing each individual's unique strengths is paramount [35]. Equations (1) and (2), as mentioned earlier, provide a methodology for identifying and calculating CIS, allowing employees to achieve optimal performance within their strength structures. However, this approach underscores an individual's relative strengths, presenting utility optimality relative to the individual. Due to the diversity of strengths among employees, direct comparisons can be complex and may lead to unconvincing ranking

results. Additionally, variations in evaluation weight vectors among employees can cause inconsistencies in evaluation criteria [36]. Considering the limitations of using CIS directly as a weight vector in evaluations, we must develop a comprehensive, credible, and fair approach that accurately reflects an individual's overall performance—one that not only underscores an employee's relative strengths but also accounts for the disparities among various strengths. A potential method involves integrating the diverse CIS and assigning appropriate weights to different characteristics. We name this method the Democratic Agent Evaluation (DAE) [37]. This method combines assessed CIS to serve as evaluation value parameters. By emphasizing individual strengths and ensuring fairness in evaluating diverse characteristics and interests, the DAE presents a scientifically robust and equitable approach [38]. It balances individualized needs and collective fairness, fostering acceptance among evaluated employees. The specific steps of this method are outlined as follows:

Step 1: The first step involves calculating the CIS for each employee, denoted as $w_i$*. This is achieved using Equation (1). By averaging the CIS of n individuals, a unified value weight structure, $w_j$*, is established.

$$w_j^* = \frac{1}{n} \sum_{i=1}^{n} w_{ij}^*, \; j = 1, 2, \cdots m. \tag{3}$$

Step 2: Following the establishment of the value weight structure, the improved ideal point model is applied for individual evaluations. This method incorporates the obtained value weight structure, predefined optimal values, and the actual performance values of individuals into the following formula to calculate the evaluation results. The key aspect of this methodology is the measurement of the distance between an individual's performance and the optimal value under the given structural parameters. A smaller distance indicates that the individual's performance is closer to the optimal, signifying superior performance.

$$g(x_k) = \sqrt{\sum_{j=1}^{m} w_j^{*2} \left( f_j^* - f_{kj} \right)^2} = \sqrt{\sum_{j=1}^{m} \left( \frac{1}{n} \sum_{j=1}^{n} w_{ij}^* \right)^2 \left( f_j^* - f_{kj} \right)^2} \tag{4}$$
$$j = 1, 2, \cdots, m; k = 1, 2, \cdots n.$$

Step 3: Finally, the evaluation results derived from the above process are organized in ascending order. This arrangement allows for a clear and structured comparison of performances among individuals, highlighting those who are closest to the optimal performance values.

### 3.2.2. Substitutability Assessment

The above section extends the ranking methodology, which is initially based on individuals' relative strengths, to also consider the overarching interests of the group. A critical distinction needs to be made between relative and absolute strengths. Relative strength is a comparative measure, referring to an individual's lateral comparison, demonstrating superior performance in a particular aspect compared to others [39]. However, it does not automatically equate to absolute strength, nor does it rule out the possibility of other employees surpassing in the same aspect. It is only when an employee, evaluated against all other members based on their CIS, still maintains a leading position that we can assert they possess a certain degree of absolute advantage [40]. Therefore, we need to further analyze the extent to which employees can be replaced within the organization in order to comprehensively assess their developmental status at a given stage. Specifically, this can be achieved through the optimization of Individual Agent Evaluation (IAE) ranking methods [37]. Due to the uniqueness of CIS, it identifies the value parameter structure that maximizes the comprehensive value for each employee within the scope of assessment. This maximum value can be considered as the employee's maximum potential within the organization. By evaluating other members against the standards of a target employee within this structure, we can determine their relative performance. If the target employee

consistently ranks at the top in their CIS evaluation, it indicates a lower likelihood of them being replaced by others. Thus, analyzing absolute advantage offers valuable insights into the degree of substitutability among employees and is also beneficial to the organization's overall planning of human resources. The specific steps for this calculation are as follows:

Step 1: Comprehensive value assessment of substituted employee: using Equation (1), we assess each employee's CIS ($w_i^*$), calculating the comprehensive value of employee $p$ within their own strengths structure.

$$g(x_p) = \sqrt{\sum_{j=1}^{m} w_p^{*2} \left( f_j^* - f_{pj} \right)^2} \quad p = 1, 2, \cdots, n \tag{5}$$

Step 2: Next, the comprehensive value of all other employees ($t$) is calculated within the CIS framework of the substitute employee ($p$). This step is crucial in understanding other members' performance when evaluated using the same criteria as the substitute employee.

$$g(x_t) = \sqrt{\sum_{j=1}^{m} w_p^{*2} \left( f_j^* - f_{tj} \right)^2}$$
$$t = 1, 2, \cdots n; \ p = 1, 2, \cdots, n; \ t \neq p \tag{6}$$

Step 3: Finally, the substitutability of each member is calculated. Our analysis shows that evaluations based on an individual's CIS do not always ensure a top-ranking position for themself. In scenarios where other members rank higher, they can be considered potential substitutes. However, it is critical to recognize that each employee has unique contributions within the group, and an individual's value should not be overlooked due to the presence of a few outstanding members. To this end, we first determine the degree to which other members can substitute for the individual in question. Subsequently, the individual's substitution rate is calculated based on the group's average substitution rate. Given that smaller values from formulas (1) and (2) are indicative of better evaluations, we normalize these inverse indicators prior to calculating the substitutability index ($v_p$).

$$v_{tp} = \frac{1/g(x_t)}{1/g(x_p)} = \frac{g(x_p)}{g(x_t)}$$
$$t = 1, 2, \cdots n; \ p = 1, 2, \cdots, n; \ t \neq p \tag{7}$$

$$v_p = \frac{1}{n-1} \sum_{t=1}^{n} v_{tp} \quad p = 1, 2, \cdots, n; \ t \neq p \tag{8}$$

Step 4: Following the aforementioned steps, we determine the substitution degree for each employee. A higher numerical value indicates a greater likelihood of being substituted by other members. The results are then arranged in ascending order, facilitating a clear understanding of substitutability within the organization.

### 3.3. Analysis of Employee Training Strategies

The application of CIS recognition technology provides two critical dimensions of employee assessment: comprehensive ranking and replaceability based on CIS. The former portrays the level that employees can achieve within the organization, taking into consideration their relative advantages and democratic opinions, while the latter reflects the absolute strength and indispensability of an employee's skill structure within the group. These dimensions aid in pinpointing employees' organizational roles, facilitating the formulation of effective development and training strategies. Employees' comprehensive rankings can be divided into high and low categories, while absolute strengths can be categorized based on replaceability (for a larger assessment pool, further subdivisions can be made). This bifurcation results in four distinct employee categories (see Table 1), each requiring tailored management strategies.

**Table 1.** Characteristics of four distinct employee categories.

| Categories | Comprehensive Strength | Advantages | Disadvantages |
| --- | --- | --- | --- |
| Higher Ranking and Lower Substitutability (HR and LS) | Strong | Outstanding | Not Evident |
| Higher Ranking and Higher Substitutability (HR and HS) | Strong | Outstanding | Not Evident |
| Lower Ranking and lower substitutability (LR and LS) | Moderate or Weak | Outstanding | Evident |
| Lower Ranking and Higher Substitutability (LR and HS) | Moderate or Weak | Not Outstanding | Evident |

HR and LS employees: These are the organization's standout performers, displaying robust competence across various domains and being challenging to replace. Their developmental focus should be on sustaining leadership positions and cultivating specialized strengths. Organizations can encourage these employees to deepen their expertise in selected areas, enhancing their potential and overall organizational talent value.

HR and HS employees: These individuals demonstrate balanced development in various fields under the democratic collective advantage, exhibiting strong overall competence but lacking distinct strengths or weaknesses. This balance makes them more susceptible to replacement. For such employees, organizations should aid in identifying and specializing in areas where they can develop unique strengths. Their goal in cultivating strengths differs from that of versatile individuals; it primarily aims to increase the difficulty in being replaced by others, thereby moving towards the direction of star employees and developing into a reserve force within the organizational talent pool.

LR and LS employees: Characterized by unique abilities in specific areas, these employees possess irreplaceable strengths but also noticeable weaknesses, leading to moderate overall competence. While they may be challenging to replace in the short term, the emergence of substitutes could undermine their value. Development plans should focus on maintaining their unique strengths while addressing their weaknesses.

LR and HS employees: These employees typically lack distinctive strengths and exhibit average performance, often fulfilling supportive roles within the team. Their lack of specialization in any area makes significant short-term improvements challenging. To increase their organizational value, they should focus on developing expertise in specific areas and addressing weaknesses.

## 4. Illustrative Examples

Environmental Technology Vanguard Limited is a high-tech enterprise founded in 2010, dedicated to advancing environmental protection and sustainable development. Specializing in the research and application of environmental technologies, the company focuses on areas such as air and water quality monitoring, waste disposal, and renewable energy. Driven by innovation and technology, Environmental Technology Vanguard has consistently delivered efficient and intelligent environmental solutions to its clients. Over the years, the company has built a highly professional and diverse team, consisting of 12 technical engineers committed to continuously enhancing their technological and innovative capabilities. Recognizing the ongoing changes in technology and the market, there is a growing need to align the skills of the team members with these evolving demands. Therefore, guiding the development of the team's personnel becomes imperative. Applying the earlier established technical approach, a team status analysis and development training plan will be implemented for the 12 team members. The evaluation criteria, partly based on external references [41], have been adapted to align with the company's specific value propositions. Consequently, a final evaluation framework has been established, consisting of three secondary and eight tertiary indicators, as detailed in Table 2.

**Table 2.** Value indicators.

| | |
|---|---|
| **Work Attitude (WA)** | Proactivity (Pro.) |
| | Responsibility (Res.) |
| **Work Performance (WP)** | Work Quality (WQ) |
| | Work Efficiency (WE) |
| | Innovative Suggestions (IS) |
| **Work Capability (WC)** | Business Capability (BC) |
| | Communication Capability (CC) |
| | Teamwork Capability (TC) |

To ensure a holistic assessment, we employed the 360-degree feedback method (alternative methods can be implemented as necessary). This involves self-assessment by the team members and anonymous scoring from colleagues, supervisors, and clients (the processing of scores and the verification of data accuracy are not the primary focuses of this study, and hence, will not be elaborated upon in detail). The assessment results for members M1–M12 across criteria 1–8, rated on a 0–10 scale, are presented in Table 3.

**Table 3.** The scores of team members.

| | WA | | WP | | | WC | | |
|---|---|---|---|---|---|---|---|---|
| | **Pro.** | **Res.** | **WQ** | **WE** | **IS** | **BC** | **CC** | **TC** |
| M1 | 7.9 | 7.7 | 6.9 | 8.8 | 6.7 | 7.9 | 5.6 | 5.3 |
| M2 | 8.0 | 8.2 | 7.5 | 8.9 | 8.7 | 7.9 | 9.0 | 8.8 |
| M3 | 5.8 | 7.4 | 7.5 | 5.2 | 5.7 | 6.3 | 6.7 | 5.9 |
| M4 | 6.0 | 6.5 | 5.1 | 7.1 | 4.8 | 9.7 | 4.5 | 7.2 |
| M5 | 8.6 | 8.9 | 7.4 | 7.0 | 8.0 | 4.9 | 7.7 | 8.4 |
| M6 | 7.2 | 7.6 | 7.6 | 7.5 | 8.6 | 7.7 | 8.5 | 8.4 |
| M7 | 7.3 | 5.5 | 5.2 | 7.5 | 6.9 | 7.8 | 7.9 | 8.0 |
| M8 | 8.7 | 8.7 | 6.9 | 6.6 | 8.0 | 5.0 | 6.2 | 8.1 |
| M9 | 7.6 | 7.8 | 6.8 | 8.4 | 6.4 | 7.7 | 5.3 | 4.9 |
| M10 | 4.3 | 7.0 | 9.8 | 8.1 | 5.8 | 5.5 | 6.6 | 6.8 |
| M11 | 8.8 | 9.2 | 7.5 | 6.9 | 8.2 | 5.7 | 7.9 | 8.6 |
| M12 | 7.4 | 5.5 | 5.7 | 7.4 | 7.8 | 8.3 | 8.0 | 8.8 |

Using the outlined calculation steps, along with Equations (1) and (2), we determined the CIS for the 12 technical engineers, as shown in Table 4.

By averaging these CIS values, we derived a unified value–weight structure. The value–weight structure is as follows: (0.115, 0.150, 0.141, 0.120, 0.091, 0.160, 0.097, 0.127). Employing Equation (4), we then calculated the individual performance values and CIS-based rankings, presented in Table 5.

Further, calculations were performed for each member's replaceability and subsequent rankings using Equations (5)–(8), with the results depicted in Table 6.

These calculations yielded comprehensive rankings and replaceability rankings for each member. The members were then categorized based on these criteria, as illustrated in Table 7.

**Table 4.** CIS of team members.

| | WA | | WP | | | WC | | |
|---|---|---|---|---|---|---|---|---|
| | Pro. | Res. | WQ | WE | IS | BC | CC | TC |
| M1 | 0.139 | 0.116 | 0.064 | 0.426 | 0.056 | 0.139 | 0.032 | 0.028 |
| M2 | 0.062 | 0.076 | 0.039 | 0.204 | 0.146 | 0.056 | 0.246 | 0.171 |
| M3 | 0.083 | 0.215 | 0.233 | 0.063 | 0.079 | 0.106 | 0.134 | 0.087 |
| M4 | 0.005 | 0.007 | 0.004 | 0.010 | 0.003 | 0.957 | 0.003 | 0.011 |
| M5 | 0.207 | 0.335 | 0.060 | 0.045 | 0.101 | 0.016 | 0.077 | 0.159 |
| M6 | 0.059 | 0.080 | 0.080 | 0.074 | 0.235 | 0.087 | 0.205 | 0.180 |
| M7 | 0.117 | 0.042 | 0.037 | 0.136 | 0.088 | 0.175 | 0.193 | 0.212 |
| M8 | 0.294 | 0.294 | 0.052 | 0.043 | 0.124 | 0.020 | 0.034 | 0.138 |
| M9 | 0.142 | 0.170 | 0.080 | 0.321 | 0.063 | 0.155 | 0.037 | 0.032 |
| M10 | 0.001 | 0.004 | 0.972 | 0.011 | 0.002 | 0.002 | 0.003 | 0.004 |
| M11 | 0.192 | 0.432 | 0.044 | 0.029 | 0.085 | 0.015 | 0.063 | 0.141 |
| M12 | 0.078 | 0.026 | 0.029 | 0.078 | 0.109 | 0.182 | 0.132 | 0.366 |

**Table 5.** Performance and ranking of team members.

| | M1 | M2 | M3 | M4 | M5 | M6 | M7 | M8 | M9 | M10 | M11 | M12 |
|---|---|---|---|---|---|---|---|---|---|---|---|---|
| Performance Value | 0.1063 | 0.0653 | 0.1310 | 0.1312 | 0.1049 | 0.0806 | 0.1187 | 0.1131 | 0.1138 | 0.1272 | 0.0930 | 0.1088 |
| Rankings | 5 | 1 | 11 | 12 | 4 | 2 | 9 | 7 | 8 | 10 | 3 | 6 |

**Table 6.** Replaceability and ranking of team members.

| | M1 | M2 | M3 | M4 | M5 | M6 | M7 | M8 | M9 | M10 | M11 | M12 |
|---|---|---|---|---|---|---|---|---|---|---|---|---|
| Replaceability | 0.6172 | 0.4265 | 1.0910 | 0.1111 | 0.5796 | 0.6299 | 0.8547 | 0.6324 | 0.7827 | 0.0664 | 0.4303 | 0.6071 |
| Rankings | 7 | 3 | 12 | 2 | 5 | 8 | 11 | 9 | 10 | 1 | 4 | 6 |

**Table 7.** Classification results of team members.

| | |
|---|---|
| **HR and LS** | M2, M11, M5, M12 |
| **HR and HS** | M6, M1 |
| **LR and LS** | M10, M4 |
| **LR and HS** | M8, M9, M7, M3 |

Our analysis identifies M2, M11, M5, and M12 as the standout performers within the team. The organization should encourage them to further develop in the direction of their relative strengths or areas of interest. For instance, M2, known for exceptional communication skills, would be well suited for roles emphasizing team coordination or maintaining client relationships. M6 and M1, while holding good organizational positions, face the risk of being replaced due to the balanced nature of their skills, particularly in the case of M6. The organization should guide M6 to develop unique, irreplaceable competencies. M10 and M4, with their specialized skills, are less replaceable in the short term. M10, for instance, excels in task quality, and M4 is notable for technical capability. However, their significant weaknesses in other areas have impacted their overall rankings. For example, M4's communication skills are notably weaker compared to others, necessitating targeted

improvement. M8, M9, M7, and M3 are currently in supportive roles within the team, lagging in both overall rankings and strength. The organization should assist these members in identifying and nurturing their relative strengths to incrementally enhance their value over time.

## 5. Summary and Conclusions

### 5.1. Conclusions

Our study, grounded in a people-centric perspective of sustainable management, introduces a decision-making framework for formulating employee-training strategies. This framework is instrumental in revitalizing human resources. It progresses through three distinct stages: Initially, it leverages technology to identify CIS, revealing the unique potential of each employee. Subsequently, it assesses employees' roles within the organization by examining their overall competence and replaceability, guided by CIS. Lastly, it applies a composite decision-making process to analyze the development phases of employees, leading to the design of tailored human resource development strategies. The efficacy of this methodology is demonstrated through illustrative examples.

### 5.2. Theoretical Implications

This study significantly enriches the theoretical discussion by offering a comprehensive methodology that centers on human resource sustainability. Moving beyond conventional qualitative research methodologies [42,43], it integrates quantitative assessment and decision-making frameworks, thereby enhancing the human resource development literature. This innovative effort yields critical insights for sustainable human resource research driven by a people-focused ideology, expanding the scope of investigation. Furthermore, this study introduces evaluation and decision-making tools designed to gauge the present condition and prospective pathways of organizational members, providing solid methodological foundations for the strategic development and planning of human resources.

### 5.3. Practical Implications

Firstly, this study pioneers a decision-making framework grounded in employee needs, acknowledging the distinct strengths and characteristics of each individual. This method honors the diversity among employees, a key factor in addressing their developmental requirements and ensuring the organization's access to a varied talent pool [44]. This aligns seamlessly with the principles of sustainable development. Secondly, this research equips organizations with insights into the capabilities, distribution of attributes, and the replaceability among staff members. By doing so, it enables the precise development of employee profiles and optimizes their inherent capabilities, thereby promoting a dynamic and reciprocal growth process between the organization and its workforce.

### 5.4. Limitations and Future Research Direction

While our study has developed a method to enhance the regeneration of human resources from a sustainable development perspective, offering an effective approach to improving organizational talent cultivation mechanisms, it is not without limitations. Primarily, although quantitative methods offer objectivity, a holistic approach to individual development necessitates the integration of subjective elements, such as personal preferences and perceptions. Future studies might leverage qualitative methodologies to deepen these initial findings with experiential evidence. Secondly, the study's reliance on the accuracy and validity of quantitative data, which may be susceptible to biases in collection and processing, could compromise the reliability of evaluation outcomes. Subsequent research is encouraged to explore sophisticated techniques in data gathering and analysis to improve data quality, thereby increasing the accuracy and trustworthiness of evaluative conclusions.

**Author Contributions:** Conceptualization, X.Z. and N.C.; methodology, X.Z.; writing—original draft preparation, N.C; writing—review and editing, N.C. and B.G.; software, C.S.; supervision,

X.Z.; project administration, B.G. All authors have read and agreed to the published version of the manuscript.

**Funding:** This research received no specific grant from any funding agency in the public, commercial or not-for-profit sectors.

**Institutional Review Board Statement:** This article followed all ethical standards for carrying out research.

**Informed Consent Statement:** Not applicable.

**Data Availability Statement:** The data that support the findings of this study are available on request from the first author.

**Acknowledgments:** The authors want to acknowledge the SUSTAINABILITY Editorial Office and all the anonymous reviewers.

**Conflicts of Interest:** The authors declare that they have no financial or personal relationships that may have inappropriately influenced them in writing this article.

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
