# Peer review of "A Method to Facilitate the Regeneration of Human Resources: A Sustainability Perspective"

_sustainability, doi:10.3390/su16041648_

Round 1

Reviewer 1 Report

Comments and Suggestions for Authors

This study is to explore a method to facilitate the regeneration of organizational human resources in terms of a sustainability perspective. The topic is timely but overall study organization is not clear and not relevant to each other. The study processes of a research are not proper. The study results are not proofed with valid test.

There are some comments to improve the quality of the study as follows:

- In the title, Organizational Human Resources is not proper. There is no such terminology of "Organizational Human Resources." Use proper words. Retitle it.

- Abstract is not clear. Provide properly an overall summary of the study. - Add a study purpose in the abstraction section and in the introduction section.

- In line 30, "human resource management" is too broad. Focus on the specific study area.

- The first paragraph between 30 and 47 is not clear. The first part, the second part and the rest are not congruent with each other. Delete between line 30 and line 37. They are not pertinent to the rest of the introduction.

- Inle 39 is not clear.

- Therefore, developing a scientifically effective method to assist employee growth is pivotal for achieving organization economic performance and sustainable development. => rewrite it since it is not true.

- organizational personnel development is not correct. Use more correct words.

- Organizational development and human resource development are totally different. Rewrite it.

- This viewpoint is supported by scholars like Lee[12]. => this sentence should be rewritten more professionally.

- Add a study purpose in the abstraction section and in the introduction section.

- Add a study methodology and data sampling in the abstraction section.

- Add study necessity and contribution in the introduction section.

- Section 2 should provide a study background. But CIS suddenly appears in section 2. Review rigorously relevant study background.

- Add some data validity explanation.

- Add some statistical aspects to analyze the results. Non-parametric statistical analysis may be used to justify the study results.

- Subsection 6.1 is not clear. Each sentence is not related to each other. Rewrite it.

- Reorganize the manuscript structure.

Reviewer 2 Report

Comments and Suggestions for Authors

This paper is intended to analyze the integration of sustainability in human resource management into a mechanism for sustainable development for the organization´s staff. It utilizes a framework known as CIS to recognize and maximize individual potential. The proposed customized training mechanism for personnel is validated through illustrative examples. The main strength of the paper is in its unique approach to developing individual potential within organizations, which matters significantly for responsible management. Although the paper has several strengths, it could also improve by focusing on weaknesses like testability of its hypotheses, methodological rigor and qualitative methods used to support quantitative approach. Nevertheless, future research could attempt to complete these missing pieces through a broader look of sustainable human resource management systems.

The manuscript provides a well-organized investigation of human resource management’s position within the theme of sustainability; thus, it is suitable for the area. The listed references seem to be fresh and relevant enough to the topic, as no signs of self-citation can be seen. As the manuscript seems scientifically solid, despite analyzing critical elements of experimental design that are appropriate for hypothesis testing being insufficiently clarified. Reproducibility of results is not fully evaluable given inssuficient methodological information. The figures and tables, for example the one describing the features of worker categories appear to be relevant as well as they clarify what source data is discussed. It would also improve interpretation by providing additional details on statistical analysis. Lastly, the conclusions reached are consistent with this evidence and reasoning in a general sense as an approach to individual training for staff.

Comments on the Quality of English Language

The quality of English Language seems acceptable; there is a clear sentence structure, and an appropriate selection of vocabulary. Nevertheless, a thorough reading by a native speaker could provide an opportunity for minor editing that improves clarity or corrects some tiny mistakes.

Round 2

Reviewer 2 Report

Comments and Suggestions for Authors

From my view point, the manuscript has beensifficiently improved for publication.